# Rotational and dilational reconstruction in transition metal dichalcogenide moiré bilayers

Madeline Van Winkle [1,9], Isaac M. Craig [1,2,3,9], Stephen Carr [4,5], Medha Dandu[2], Karen C. Bustillo [2], Jim Ciston [2], Colin Ophus[2], Takashi Taniguchi [6], Kenji Watanabe [7], Archana Raja [2], Sinéad M. Griffin [2,3] & D. Kwabena Bediako [1,8] ✉

Lattice reconstruction and corresponding strain accumulation plays a key role in defining the electronic structure of two-dimensional moiré super-lattices, including those of transition metal dichalcogenides (TMDs). Imaging of TMD moirés has so far provided a qualitative understanding of this relaxation process in terms of interlayer stacking energy, while models of the underlying deformation mechanisms have relied on simulations. Here, we use interferometric four-dimensional scanning transmission electron microscopy to quantitatively map the mechanical deformations through which reconstruction occurs in small-angle twisted bilayer $MoS_2$ and $WSe_2/MoS_2$ heterobilayers. We provide direct evidence that local rotations govern relaxation for twisted homobilayers, while local dilations are prominent in heterobilayers possessing a sufficiently large lattice mismatch. Encapsulation of the moiré layers in hBN further localizes and enhances these in-plane reconstruction pathways by suppressing out-of-plane corrugation. We also find that extrinsic uniaxial heterostrain, which introduces a lattice constant difference in twisted homobilayers, leads to accumulation and redistribution of reconstruction strain, demonstrating another route to modify the moiré potential.

Moiré superlattices, formed by vertically stacking van der Waals layers with a small rotational offset and/or lattice mismatch, are a versatile platform for modulating the physicochemical behavior of two-dimensional solids[1–5]. Moiré architectures comprised of semiconducting transition metal dichalcogenides (TMDs) are of considerable fundamental and technological interest because they exhibit distinctively tunable optoelectronic features, such as inter- and intralayer moiré excitons and trions[2,3,6–14], as well as relatively robust correlated electronic phases[15–21]. While the electronic band structures of moiré superlattices can be intentionally modified by changing the extent of crystallographic misalignment between constituent layers, intrinsic lattice reconstruction also plays an underlying yet substantial role in controlling the emergent behavior in these systems[22–27]. Reconstruction of the superlattice and the development of intralayer

[1]Department of Chemistry, University of California, Berkeley, CA 94720, USA. [2]Molecular Foundry, Lawrence Berkeley National Laboratory, Berkeley, CA 94720, USA. [3]Materials Sciences Division, Lawrence Berkeley National Laboratory, Berkeley, CA 94720, USA. [4]Department of Physics, Brown University, Providence, RI 02912, USA. [5]Brown Theoretical Physics Center, Brown University, Providence, RI 02912, USA. [6]International Center for Materials Nanoarchitectonics, National Institute for Materials Science, 1-1 Namiki, Tsukuba 305-0044, Japan. [7]Research for Functional Materials, National Institute for Materials Science, 1-1 Namiki, Tsukuba 305-0044, Japan. [8]Chemical Sciences Division, Lawrence Berkeley National Laboratory, Berkeley, CA 94720, USA. [9]These authors contributed equally: Madeline Van Winkle, Isaac M. Craig. ✉e-mail: bediako@berkeley.edu

shear strain subsequently lead to reconstruction of the electronic band structure, affecting features such as the depth of the moiré potential[28], the 'flatness' of low-energy bands[29,30], and the real-space localization of charge carriers[31–33].

Group VI (Mo- and W-based) H-phase TMDs, which are non-centrosymmetric in the monolayer limit, have two distinct reconstructed forms with unique band structures depending on whether there is a parallel (P, 3R-like, near 0°) or anti-parallel (AP, 2H-like, near 60°) orientation between layers. Scanning probe[24,26,28] and electron microscopy[23] techniques have provided a qualitative picture of reconstruction in TMD moiré homo- and heterobilayers, understood in terms of the variation in interlayer stacking energy throughout the superlattice. However, the physical mechanisms by which reconstruction occurs have thus far only been simulated. Here we use Bragg interferometry[25,34], an imaging methodology based on four-dimensional scanning transmission electron microscopy[35] (4D-STEM), to directly map the intralayer mechanical deformations driving reconstruction in TMD moiré systems. We identify distinct reconstruction mechanisms for moiré homobilayers versus heterobilayers and examine their twist angle dependence, distinguishing the relative importance of local lattice rotations and dilations in both systems as well as the critical role that encapsulation layers play in affecting the balance between these in-plane deformations and out-of-plane corrugations. We also measure reconstruction-induced strain fields and demonstrate how application of an external mechanical force, e.g. heterostrain, can be leveraged to manipulate strain distributions.

## Results and discussion
### Interlayer displacement mapping

We prepared $MoS_2$ moiré homobilayers and $WSe_2/MoS_2$ moiré heterobilayers that were capped with thin (5-10 nm) hexagonal boron nitride (hBN). Twisted homobilayers were fabricated using the tear-and-stack method[36] to introduce a desired interlayer moiré twist angle ($\theta_m$). Heterobilayers were made by stacking two separate TMD monolayers with straight flake edges aligned, and the stacking orientation (P or AP) was confirmed using second harmonic generation spectroscopy (see sample fabrication details in Supplementary Note 1)[37].

To image the moiré superlattice structures, we performed 4D-STEM Bragg interferometry, described in Fig. 1a. The guiding principle of this imaging technique is that electron waves diffracted by the two TMD layers interfere with one another; in reciprocal space this leads to a modulation of the intensity measured in the overlapping regions of the TMD Bragg disks that depends on the local stacking sequence. Given a certain convergence angle of the incident beam, the amount of Bragg disk overlap is controlled by moiré twist angle ($\theta_m$) for homobilayers (Fig. 1b) and both twist angle and lattice constant percent difference ($\delta$) for heterobilayers (Fig. 1c).

The intensities in the regions of Bragg disk overlap can be used to determine the average interlayer atomic displacement vector at each beam position, providing structural information about the superlattice as recently demonstrated for twisted bilayer graphene[25]. For non-centrosymmetric systems and heterobilayers, which do not necessarily possess centrosymmetry, our methodology requires an expansion of

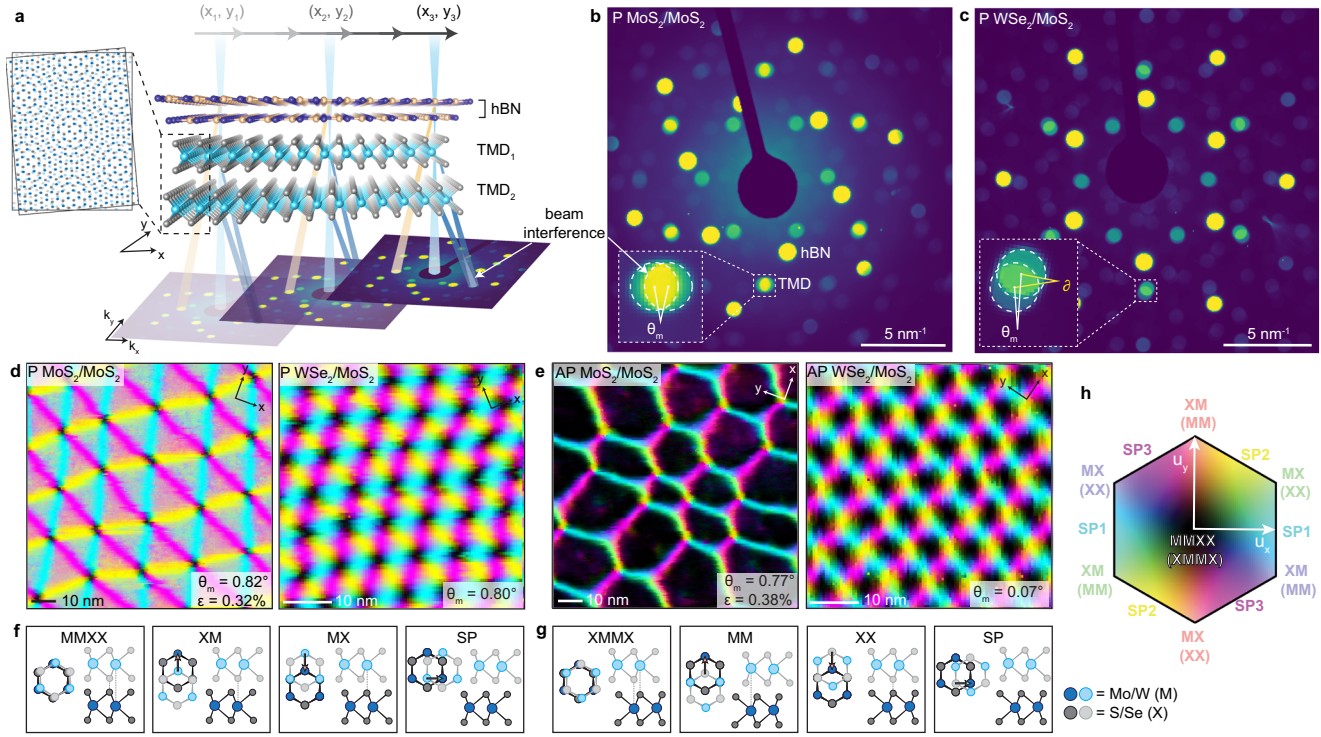

**Fig. 1 | Bragg interferometry and displacement field maps. a** Schematic of Bragg interferometry measurement. Here $(x_1, y_1)$, $(x_2, y_2)$, $(x_3, y_3)$ refer to electron beam positions. $TMD_1$ and $TMD_2$ are $MoS_2$ for homobilayers or $MoS_2$ and $WSe_2$ for heterobilayers. Dashed box and corresponding inset illustrate formation of a moiré superlattice between $TMD_1$ and $TMD_2$. Axis labels $x$, $y$ and $k_x$, $k_y$ indicate real space and reciprocal space coordinate systems, respectively. **b, c** Average convergent beam electron diffraction patterns for a $MoS_2/MoS_2$ moiré homobilayer and $WSe_2/MoS_2$ moiré heterobilayer, respectively, with moiré twist angle $\theta_m$ and lattice constant percent difference $\delta$. Overlapping TMD Bragg disks are highlighted in the insets. **d, e** Representative displacement maps for moiré bilayers with parallel (P) and anti-parallel (AP) orientations, respectively. The moiré twist angle and heterostrain are labelled as $\theta_m$ and $\epsilon$. **f, g** High-symmetry stacking sequences and corresponding displacement vectors for P and AP H-phase TMD moiré bilayers. Saddle points, or domain walls, abbreviated as SP. Metal and chalcogen denoted as M and X, respectively. **h** 2D displacement hexagon legend for the displacement field maps in **d** and **e**, signifying the magnitudes and directions of the local displacement vectors in pixel hues and values, respectively. Here $u_x$ and $u_y$ represent interlayer displacements in the $x$ and $y$ directions. $SP_1$, $SP_2$, and $SP_3$ represent the three unique saddle point stacking directions.

the fitting function used previously[25] to relate $I_j$, the overlap intensity in the $j^{th}$ Bragg disk pair, and $\mathbf{u} = (u_x, u_y)$, the local displacement vector that describes the interlayer offset between the TMD lattices (see Supplementary Note 2 for details on the full derivation of the expanded fitting function):

$$I_j = A_j \cos^2(\pi \mathbf{g}_j \cdot \mathbf{u}) + B_j \cos(\pi \mathbf{g}_j \cdot \mathbf{u}) \sin(\pi \mathbf{g}_j \cdot \mathbf{u}) + C_j \qquad (1)$$

Notably, although the hBN encapsulation layers are colocalized with the TMD layers in real space, the hBN Bragg disks are sufficiently offset from those of the TMD layers in reciprocal space and thus do not impede the structural analysis in Bragg interferometry. This 4D-STEM approach is therefore not restricted by buried interfaces, as in the case of scanning probe methods (which often require the sample surface to be exposed) and conventional real-space electron imaging methods (that can be obscured by encapsulating layers).

Example displacement maps for P and AP moiré bilayers are provided in Fig. 1d, e. For the P case, the high-symmetry stacking orders include MMXX, XM, MX, and SP (saddle point), as described in Fig. 1f. In comparison, the stacking orders in the AP case include XMMX, MM, XX, and SP, shown in Fig. 1g. It is of particular note that in Fig. 1d, e, each pixel color encodes quantitative information about the local displacement vector (illustrated as arrows in Fig. 1f, g) within the displacement zone depicted in Fig. 1h. We observe sharp triangular features in the displacement map for the P moiré homobilayer and a hexagonal structure for the AP orientation, similar to what has been previously reported[23,24]. The moiré pattern is much smaller for the heterobilayer cases, considering the maximum possible periodicity is $\approx 8$ nm for $\delta_{MoS_2/WSe_2} = 3.96\%$. Notably, while the heterobilayer displacement fields appear more like that of a rigid moiré lattice (see Supplementary Fig. 5), ostensibly suggesting that reconstruction is relatively weak compared to the twisted homobilayers, the strain mapping and geometric analyses that follow show that reconstruction remains strong even in these cases.

## Rotational reconstruction in homobilayers

While 4D-STEM-based strain mapping has typically been performed by tracking changes in Bragg disk positions throughout a data set[38,39], accurate registration of disk positions is very challenging for moiré systems where diffraction disks from the constituent monolayers have substantial overlap and a relatively low signal-to-noise ratio. As an alternative, by taking the gradient of the displacement vector field, we can calculate the intralayer 2D strain tensor at each position in the sample[25,40–42]. From the strain tensor, we then derive information about local intralayer fixed-body rotations and deformations in the moiré bilayer (see Methods), which provides insight into the reconstruction mechanisms in these systems.

First we will consider the $MoS_2$ moiré homobilayers. The intralayer reconstruction rotation ($\theta_R$), shown in Fig. 2a–c, indicates the difference between the pre-imposed interlayer moiré twist angle ($\theta_m$) and the measured total fixed-body rotation ($\theta_T$) in each TMD layer: $\theta_R = \theta_T - (\theta_m/2)$. For the P orientation (Fig. 2a) we observe a reconstruction rotation field that is reminiscent of the triangular rotation field that has been observed in twisted bilayer graphene[25]. By plotting the average reconstruction rotation as a function of interlayer displacement ($u_x, u_y$) (insets in Fig. 2a–c), we observe that regions with the highest calculated stacking energy (MMXX, 59.1 meV/M vs. XM/MX[23,43], Supplementary Fig. 6) have $\theta_R > 0°$, which increases the local total rotation, $\theta_T^{MMXX}$, and consequently shrinks the MMXX stacking domain, while regions with low stacking energy (XM and MX) have $\theta_R < 0°$ and expand into commensurate triangular domains. For AP moiré homobilayers, a similar principle applies, but the calculated relative energies of the various high-symmetry stacking orders present changes. Here, XX regions have the highest stacking energy

(58.8 meV/M vs XMMX[23,43]), thus $\theta_R > 0°$, while XMMX regions have the lowest stacking energy and $\theta_R < 0°$ (Fig. 2b, c).

To determine whether other deformation mechanisms contribute to the reconstruction process, we also calculated dilations from the measured local strain tensors. Dilation, also referred to as dilatation or an in-plane volumetric strain[40–42], describes the local change in volume relative to that of the rigid moiré for a given $\theta_m$. In the case of moiré bilayers the dilation effectively represents the change in the relative lattice constants of the two TMD layers (that is, a 1% dilation implies a 1% increase in the lattice constant difference, thereby shrinking the local stacking domain). Figure 2d–f show the average dilation as a function of interlayer displacement. In contrast to the reconstruction rotations, we do not measure any systematic trends in dilation based on the local stacking order for the moiré homobilayers (both P and AP cases). These data reveal that intralayer volumetric strains do not significantly contribute to reconstruction of the homobilayer lattices. Instead, variations in the distribution of local fixed-body rotations drive the reconstruction and are responsible for the stark morphological differences between reconstructed P and AP moiré homobilayers. This result may be intuitively rationalized considering the lattice mismatch in moiré homobilayers arises almost entirely from rotational misalignment rather than a lattice constant difference for samples measured (see Supplementary Note 6 for discussion on heterostrain considerations).

These rotation-driven lattice reconstruction mechanisms would be expected to generate intralayer strain as an inherent property of the moiré superlattice[25,44,45]. These inhomogeneous, intrinsic, intralayer strain fields are thought to be particularly important in AP moiré homobilayers, where they are implicated in the tight confinement of charge carriers in XX and MM sites and subsequent formation of ultraflat electronic bands[29,32]. In the case of rotation-driven reconstruction, shear is the predominant type of strain present and has been theoretically predicted to occur at the boundaries between stacking domains[29]. To visualize and measure the reconstruction strain fields, we calculated the engineering shear strain ($\gamma_{max}$, also called principal shear) at each point in the moiré superlattice. This $\gamma_{max}$ value indicates the maximum amount of intralayer shear strain present in any direction[40–42]. Indeed, we measure a concentration of shear strain in the saddle point (SP) areas (i.e., soliton/domain walls) between regions with the same sign of $\theta_R$ (Fig. 2g–i). The measured shear strain fields align closely with those we obtain from simulations using a rotational reconstruction field (see Supplementary Note 7), corroborating the assertion that local rotations are the dominant type of mechanical deformation in TMD moiré homobilayers, spontaneously generating these intralayer shear strain fields. Schematics depicting the rotation-driven reconstruction model and accumulation of shear strain at domain boundaries are provided in Fig. 2j.

Figure 2k–n show that the relative stacking area and average $\theta_R$ vary with twist angle for each type of high-symmetry stacking order. Notably, there are two regimes of reconstruction observed for the AP moiré homobilayers. While $\theta_R^{XMMX}$ is consistently $< 0°$ and $\theta_R^{XX} > 0°$ for all $\theta_m$ measured, $\theta_R^{MM}$ switches sign at a critical twist angle ($\theta_c$) around 1.25–1.5° (Fig. 2k). Although MM regions are higher energy than XMMX regions for group VI TMDs, the difference is relatively small (13.8 meV/M vs. XMMX[23,43]). As a result, at larger $\theta_m$ ($> \theta_c$) we observe a slightly negative $\theta_R$ in the MM regions, allowing some local domain expansion (Fig. 2l). Meanwhile as $\theta_m$ decreases, MM regions instead shrink ($\theta_R > 0°$) to accommodate rapid expansion of XMMX into large hexagonal domains (Fig. 2k, l). On the other hand, only one reconstruction regime was observed for the P homobilayers over the range of $\theta_m$ measured (Fig. 2m, n).

Comparing how reconstruction evolves with twist angle in the P and AP moiré homobilayers, it appears that the P orientation is more strongly reconstructed overall. When $\theta_m \approx 2°$, the relative stacking area of the MMXX regions is 36% of that expected in a rigid P moiré, while

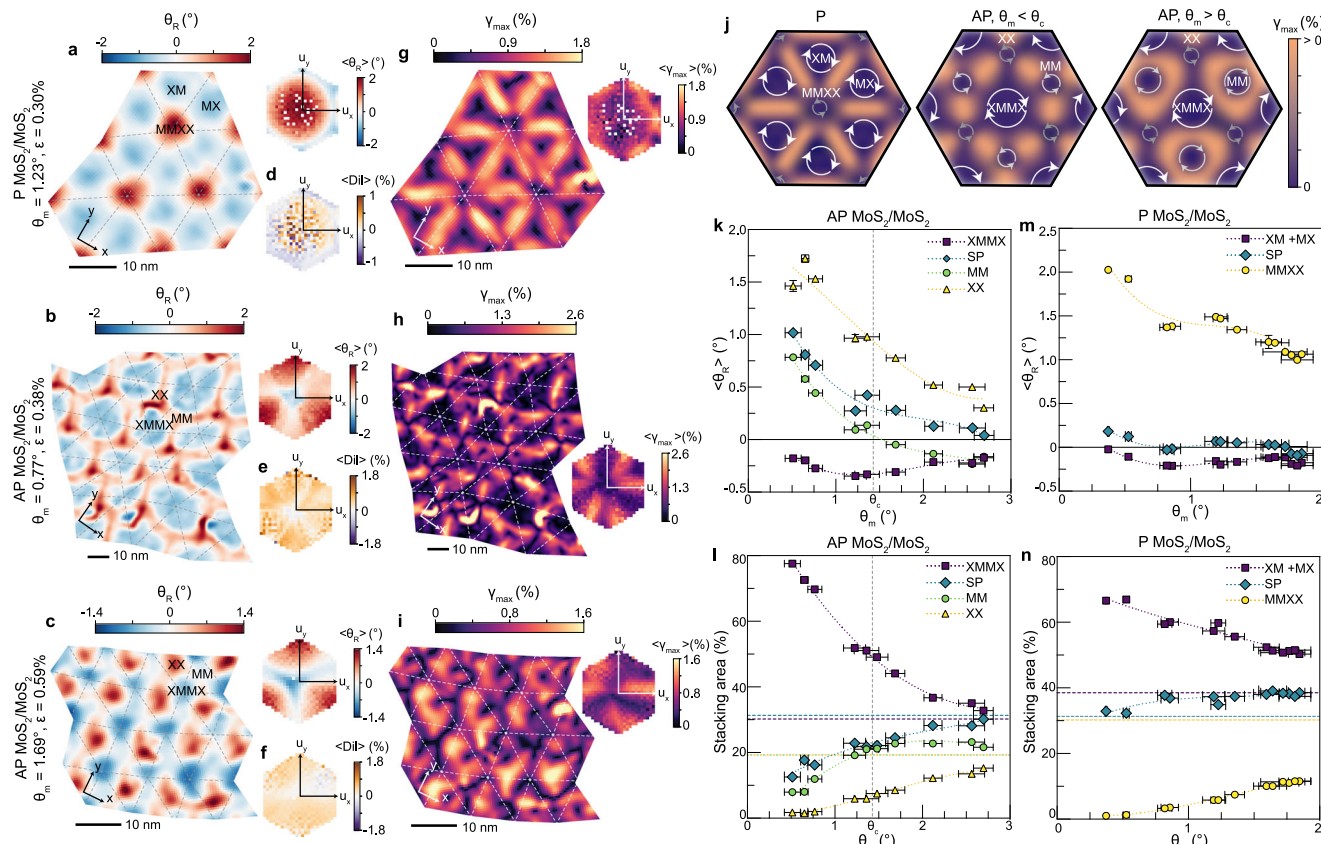

**Fig. 2 | Strain fields and area fractions in twisted homobilayers. a–c** Maps of local reconstruction rotation ($\theta_R$) and (**g–i**) maximum shear strain ($\gamma_{max}$) for P-stacked MoS$_2$/MoS$_2$ with $\theta_m = 1.23°$ and AP-stacked MoS$_2$/MoS$_2$ with $\theta_m = 0.77°$ and $1.69°$. $\varepsilon$ indicates average heterostrain. Hexagonal insets show the average reconstruction rotation ($\langle\theta_R\rangle$) and shear strain ($\langle\gamma_{max}\rangle$) as a function of interlayer displacement ($u_x, u_y$). The overlaid dashed lines correspond to the moiré unit cell geometry, determined from the displacement maps. **d–f** Average dilations ($\langle Dil\rangle$) as a function of displacement. **j** Rotation-driven reconstruction schematics for a P-stacked and two AP-stacked moiré homobilayer cases ($\theta_m < \theta_c$ and $\theta_m > \theta_c$, where $\theta_c$ is the critical twist angle separating two reconstruction regimes). Yellow indicates accumulation of shear strain ($\gamma_{max} > 0$) and blue indicates no shear strain. Arrows illustrate the measured direction of $\theta_R$, with counterclockwise rotation defined as $\theta_R > 0$. Arrow sizes depict relative growth or shrinkage of local stacking domain, not drawn to scale. **k, m** Average reconstruction rotations and **l, n** relative stacking areas as a function of twist angle ($\theta_m$) for AP and P stacking orientations. Horizontal dashed lines in (**l, n**) indicate the relative stacking areas in a rigid moiré based on the chosen partitioning of the displacement space, see Methods. In **k–n**, horizontal error bars represent standard deviations and vertical error bars represent standard errors. Dotted trend lines are polynomial fits to the experimental data, included as visual guides.

the relative stacking area of the XX regions is 79% of that in a rigid AP moiré, indicating that the AP structure is nearing the rigid case by that point but the P structure is still markedly relaxed, in line with what has been theoretically computed previously[23]. These results point to the fact that, although the P and AP moiré configurations have similar ranges of interlayer stacking energies (Supplementary Fig. 6)[23,43], fewer stacking sequences in the AP case correspond to the energy extremes. While reconstruction in a P moiré bilayer is driven by a preference for both MX and XM stacking over MMXX, reconstruction in AP moiré bilayers is driven predominantly by a preference for XMMX over XX stacking with relatively little preference for size of MM regions. This produces a stronger driving force for reconstruction in the P moiré bilayer.

### Dilational reconstruction in heterobilayers

We next turn our attention to the moiré heterobilayers, composed of WSe$_2$ and MoS$_2$. In these systems, the lattice constant difference between the two dissimilar TMD layers, rather than (or in addition to) global interlayer rotation, generates the moiré pattern. Based on the displacement fields shown in Fig. 1d, e, it initially appears as though there is minimal lattice reconstruction in the heterobilayers. However, comparing the relative areas of the different stacking sequences to those expected for a rigid moiré superlattice, it becomes evident that there is an overall expansion of lower-energy XM/MX (P) and XMMX (AP) regions and contraction of higher-energy MMXX (P) and XX (AP) regions (Fig. 3a–c), signifying that reconstruction has indeed occurred.

Real-space local dilation maps and corresponding average dilations as a function of displacement vector are provided for both AP (Fig. 3d, e) and P (Fig. 3f) heterobilayer cases. Unlike moiré homobilayers, these heterobilayers show prominent stacking order-dependent dilation patterns. Namely, there are positive dilations in the XX (AP) and MMXX (P) regions and negative dilations in XMMX (AP) and XM/MX (P) regions. There are relatively small dilations in the MM regions of the AP superlattice due to a decreased preference for the size of these domains, similar to what was observed in the AP twisted homobilayers in Fig. 2k, l. Altogether, these measurements show that the lattice constant mismatch decreases (increases) in domains with low (high) stacking energy to cause local volumetric expansion (contraction) of these domains, as depicted in the schematics in Fig. 3g.

Next we consider the effects of introducing an interlayer twist angle on heterobilayer reconstruction mechanisms. While the average local reconstruction rotations are relatively weak and independent of stacking order for the AP heterobilayer with a near zero-degree twist angle (Fig. 3h, $\theta_m = 0.13°$), these rotations strengthen in the AP heterobilayer with a larger interlayer twist (Fig. 3i, $\theta_m = 1.07°$) and their distribution resembles that of the AP twisted homobilayer (Fig. 2c). Thus, moiré heterobilayers can host a combination of rotational and

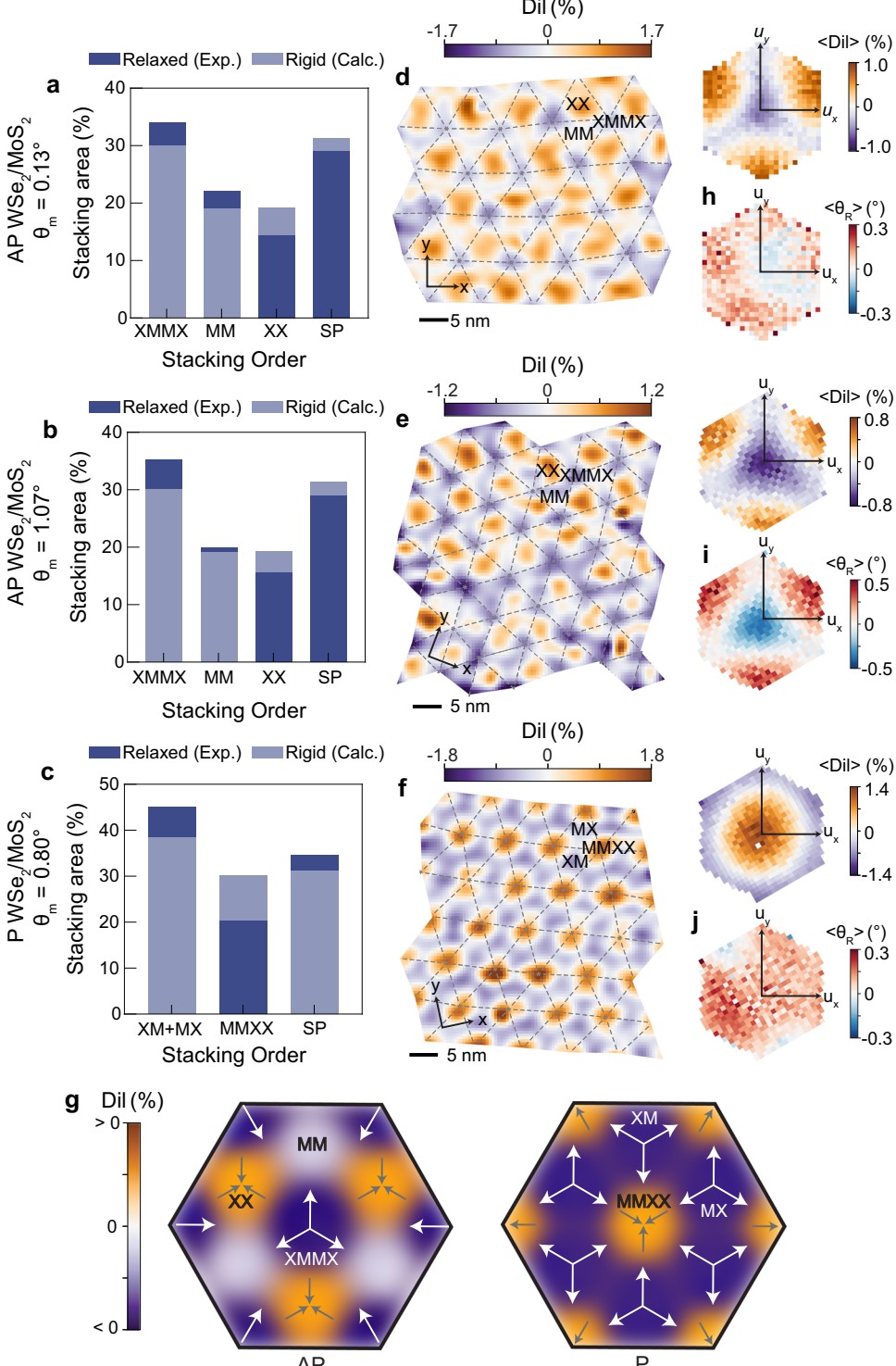

**Fig. 3 | Strain fields and area fractions in heterobilayer moirés.** Relative stacking area percentages for (**a**) AP ($\theta_m = 0.13°$), (**b**) AP ($\theta_m = 1.07°$), and (**c**) P ($\theta_m = 0.80°$) WSe$_2$/MoS$_2$. Light blue-gray bars indicate stacking areas calculated for the rigid (not reconstructed) moiré, and dark blue bars represent experimentally measured values. **d**–**f** Maps of local dilations for the samples from **a**–**c**. Hexagonal insets show the average dilation ($\langle Dil \rangle$) as a function of interlayer displacement ($u_x, u_y$). The overlaid dashed lines correspond to the moiré unit cell geometry, determined from the displacement maps. **h**–**j** Corresponding 2D plots of average reconstruction rotation ($\langle \theta_R \rangle$) as a function of displacement. **g** Schematics of dilation-driven reconstruction and accumulation of volumetric strain. Orange indicates a positive dilation and purple indicates a negative dilation. Arrows illustrate volumetric expansion (pointing outward) or compression (pointing inward). Arrow sizes represent relative growth or shrinkage of the local stacking domain, not drawn to scale.

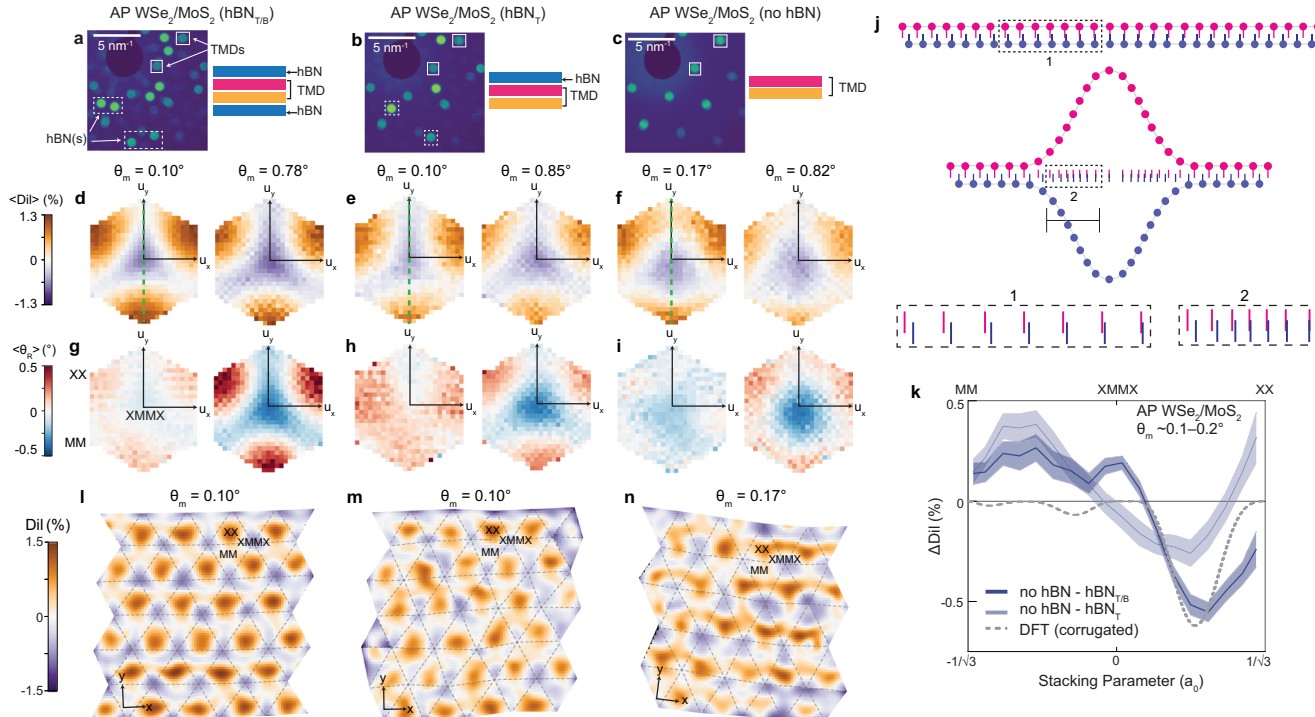

**Fig. 4 | Influence of encapsulation layers on heterobilayer reconstruction.**
**a–c** Example convergent beam electron diffraction patterns for an AP WSe$_2$/MoS$_2$ sample with three regions: fully encapsulated with hBN (**a**, $\theta_m = 0.78°$), encapsulated on one side with hBN (**b**, $\theta_m = 0.85°$), and freely suspended (**c**, $\theta_m = 0.82°$). Dashed and solid boxes highlight representative hBN and TMD diffraction disks, respectively. **d–f** Average dilations ($\langle Dil \rangle$) as a function of interlayer displacement ($u_x, u_y$) for fully capped (**d**, $\theta_m = 0.10°, 0.78°$), partially capped (**e**, $\theta_m = 0.10°, 0.85°$), and suspended (**f**, $\theta_m = 0.17°, 0.82°$) AP WSe$_2$/MoS$_2$ samples. **g–i** Corresponding 2D plots of average reconstruction rotation ($\langle \theta_R \rangle$) as a function of displacement. **j** Schematic (exaggerated) depicting the effects of out-of-plane corrugation on projected interlayer displacements. Magnified pictures of select regions are

provided in boxes 1 and 2). **k** Difference in dilation ($\Delta Dil$) as a function of stacking parameter for suspended versus fully encapsulated (dark blue-gray curve) and suspended versus partially encapsulated (light blue-gray curve) structures with $\theta_m \approx 0.1–0.2°$. Dilation values for each case obtained by taking line cuts through the average dilation plots in **d–f**, shown as green dashed lines. Shading indicates standard error. Gray dashed line represents the theoretically calculated $\Delta Dil$ comparing the corrugated versus rigid heterobilayer. **l–n** Maps of local dilations for fully capped (**l**, $\theta_m = 0.10°$), partially capped (**m**, $\theta_m = 0.10°$), and suspended (**n**, $\theta_m = 0.17°$) AP WSe$_2$/MoS$_2$. The overlaid dashed lines correspond to the moiré unit cell geometry, determined from the displacement maps.

dilational relaxation given that the imposed moiré twist angle is sufficiently large. Interestingly, we do not observe stacking-dependent rotations in the P heterobilayer with non-zero twist (Fig. 3j, $\theta_m = 0.80°$). This could be due to a couple of factors. First, the twist angle in the sample measured might be below the threshold for substantial contributions from rotational relaxation. A second possibility is that out-of-plane corrugations in the layers have weakened and delocalized the rotational reconstruction, as we will discuss next. Simulated dilation and rotation fields for P and AP heterobilayers are provided in Supplementary Figs. 10–12.

**Effects of encapsulation layers on reconstruction**
Theoretical calculations and scanning tunneling microscopy topography measurements have suggested that heterobilayer systems can relax through out-of-plane corrugations, where there is a stacking order-dependent variation in the interlayer spacing[26,28,46]. Such corrugations have been invoked as critical to relaxation and the resulting electronic properties. However, to the best of our knowledge, there have been no experimental studies that directly compare reconstruction in encapsulated and suspended structures. To investigate this point, we prepared WSe$_2$/MoS$_2$ heterobilayers with three regions: fully encapsulated (hBN on top and bottom), partially encapsulated (hBN on top), and suspended (no hBN). Example convergent beam electron diffraction patterns from the three regions with varying extents of encapsulation are provided in Fig. 4a–c for an AP sample with a moiré twist angle near 0.8°. Figure 4d–f, g–i show the corresponding average reconstruction dilations and rotations, respectively,

for these three regions in both the 0.8° sample and a similar sample with $\theta_m \approx 0.1–0.2°$. In these 2D plots, it is evident that hBN encapsulation layers affect the magnitude and extent of localization of the reconstruction rotation and dilation fields for both twist angle ranges (real-space maps provided in Supplementary Figs. 15,16). Specifically, the fully encapsulated regions show the strongest, most localized dilations and rotations, while these deformations are weakest and most delocalized in the fully suspended regions. These observations suggest that hBN encapsulation suppresses out-of-plane relaxation, in turn enhancing in-plane reconstruction pathways.

To verify the hypothesis that hBN suppresses out-of-plane corrugation, it is important to first consider the effects that such a corrugation would have on the measured in-plane projections of the displacement vectors. A pictorial representation of this scenario is shown in Fig. 4j and further mathematical details are provided in Supplementary Note 9. In Fig. 4j, pink and blue vertical lines represent the projected positions of the atoms in layers 1 and 2, respectively, of an exemplar heterobilayer system with an interlayer lattice constant mismatch. The projected distance between an atom in layer 1 to its nearest neighbor in layer 2 reflects the measured local displacement vector. Based on this picture, it is clear that out-of-plane bending of the layers reduces both the apparent lattice constant of each layer and the magnitude of the interlayer displacement vectors in the projection. This effect is most pronounced in the boundary region between two high-symmetry stacking orders where the interlayer spacing is changing most rapidly (highlighted in boxes 1 and 2 in Fig. 4j). Ultimately, this corrugation produces a perceived reduction in the interlayer

lattice mismatch, analogous to a negative dilation, even in the absence of any actual changes in intralayer bond lengths. With this conceptual framework in mind, we calculate the theoretical apparent dilations as a function of stacking parameter for a corrugated AP-stacked WSe$_2$/MoS$_2$ using interlayer spacing variations determined from DFT (Supplementary Notes 7, 9). The results of this calculation are plotted as the gray dashed line in Fig. 4k.

We then quantitatively compare the dilations in the three regions of the sample with $\theta_m \approx 0.1–0.2°$, a twist angle range where rotational reconstruction is minimal and can effectively be ignored. To do so, we take line cuts through the average dilation plots (shown as green dashed lines in Fig. 4d–f) and then calculate the difference in measured dilation ($\Delta Dil$) between both the suspended and encapsulated (*no hBN* − *hBN$_{T/B}$*) and suspended and partially encapsulated (*no hBN* − *hBN$_T$*) regions as a function of stacking parameter, as shown in Fig. 4k. The experimental $\Delta Dil$ profile for the encapsulated versus suspended case displays clear similarities to that for the theoretical corrugated structure, indicating that the suspended heterobilayer has undergone a combination of in-plane and out-of-plane reconstruction while the fully encapsulated structure is reconstructed almost entirely in-plane. The residual differences between the experimental and theoretical curves suggest that some out-of-plane corrugation may remain in the encapsulated structure since the thin hBN is not perfectly rigid; however, overall the corrugations have been largely suppressed by the presence of hBN on both sides. Meanwhile, the $\Delta Dil$ profile for the partially encapsulated versus suspended case differs more substantially from that of the corrugated model due to further mixing of in-plane and out-of-plane reconstruction when only one side of the heterobilayer is encapsulated.

Taken together, these results show that hBN encapsulation layers markedly affect the balance between in-plane rotational and dilational reconstruction and out-of-plane corrugation. In addition, it is directly established that a considerable portion of the dilations we observe in the heterobilayer systems are from actual local lattice stretching and compressing in the constituent monolayers, rather than corrugation alone, which has until now only been theoretically predicted or assumed to occur[6,26,28]. As a further point, we note that full encapsulation of the moiré layers dramatically increases the homogeneity of the reconstructed superlattice, as demonstrated by comparing the real-space dilation maps for the 0.1–0.2° sample (Fig. 4l–n). The thickness of the hBN used in fabrication may also allow for some control over the extent of corrugation; thicker hBN slabs would be more rigid and should frustrate corrugations more (thus augmenting in-plane dilations and overall disorder in the sample) in comparison to thinner hBN. Such encapsulation effects may therefore be used to further tune the physics of TMD moirés. Although these encapsulation studies were only performed for the AP heterobilayer, the trends observed should be applicable to both the P heterobilayer and P/AP twisted homobilayer cases. For example, the reconstruction rotations measured for P/AP twisted bilayer MoS$_2$ in Fig. 2, which were measured for partially encapsulated structures, are likely systematically smaller than those in analogous fully encapsulated structures.

## Heterostrain effects

Moiré heterobilayers are prepared using two dissimilar TMD materials. However, introduction of a heterostrain, wherein one TMD layer is stretched relative to the other, also creates a lattice constant difference in moiré homobilayers, making them heterobilayer-like. This raises the question of how rotational and dilational reconstruction compete in heterostrained moiré homobilayers. To investigate these relaxation dynamics, we performed Bragg interferometry on P and AP moiré homobilayers with varying amounts of uniaxial heterostrain. Figure 5a–c,g–i show the average $\theta_R$, dilation, and $\gamma_{max}$ as a function of $u_x$ and $u_y$. The results show that increasing heterostrain up to typical values of $\approx 1.4\%$ does not substantially increase the prevalence of

dilational reconstruction in twisted homobilayers. Instead, local rotations remain overwhelmingly dominant in governing relaxation. The effective lattice constant mismatch in a heterostrained moiré homobilayer can be calculated using the expression $\delta = (\epsilon - \rho\epsilon)/2$, where $\delta$ is the lattice mismatch, $\epsilon$ is the percent heterostrain, and $\rho$ is the Poisson ratio (0.234 for MoS$_2$) (see Supplementary Note 5 for details). The effective mismatch for the samples studied is at most 0.5% (for $\epsilon = 1.4\%$), nearly an order of magnitude smaller than that of the WSe$_2$/MoS$_2$ system discussed in Figs. 3, 4. Thus, while there is a lattice constant difference present owing to heterostrain, our measurements reveal that the resulting mismatch is not large enough to induce substantial dilational reconstruction. These results suggest that similar reconstruction mechanisms may therefore be expected in moiré heterobilayers with a small lattice constant percent difference (e.g. WSe$_2$/MoSe$_2$, $\delta = 0.3\%$).

Instead of inducing dilational reconstruction, our strain mapping reveals that the primary effect of heterostrain is the reorganization of the intralayer shear strain fields arising from rotational reconstruction. So although heterostrain is applied uniformly, it is anisotropic in its manifestation; it is localized in the SP regions and both amplifies and distorts the existing shear strain. In addition, the primary direction of the heterostrain has significance. Figure 5d–f, j–l show $\gamma_{max}$ for P and AP moiré homobilayers with the heterostrain axis stretching versus contracting the moiré unit cell. Each of these scenarios yields a distinct strain field, including stripe (Fig. 5e, l), square (Fig. 5f), and triangular patterns (Fig. 5k). In all cases, reconstruction rotations strongly persist and shear strain continues to concentrate in the domain walls (Fig. 5a–c, g–i), demonstrating that it is thermodynamically preferable to continue reducing interlayer energy by accumulating and shifting the distribution of elastic energy (see Supplementary Note 7 for simulations).

In summary, here we have introduced the Bragg interferometry methodology for imaging non-centrosymmetric and heterobilayer systems, enabling us to directly probe intralayer mechanical deformations in TMD moiré bilayers. We demonstrate that it is variations in the symmetry of fixed-body rotation fields that are responsible for the morphological differences that have previously been observed in relaxed P and AP moiré homobilayers. The presence of an extrinsic heterostrain preserves these rotation fields and can be used to redistribute intralayer shear strain that is localized in domain boundaries, yielding a diversity of strain patterns. Since intralayer strain affects the positions of the conduction and valence band edges throughout the moiré unit cell[28] and can generate an in-plane piezopotential[31,32], manipulating the arrangement and magnitude of reconstruction strain via extrinsic application of uniaxial heterostrain should have important implications for changing the moiré potential landscape and localization of charge carriers. For example, 1D moiré potentials have led to linearly polarized exciton emission in uniaxially strained heterobilayers[47]. In addition, we find that when the lattice constant mismatch is further increased, periodic dilation patterns become the primary route through which reconstruction occurs, though contributions from local rotations are also present as the interlayer twist angle increases. The twist angle and lattice mismatch-dependent reconstruction trends we observed should be widely applicable to moiré bilayers comprised of other TMDs (e.g., H-phase MoTe$_2$, WS$_2$, etc.) and even magnetic 2D moiré superlattices consisting of CrI$_3$ bilayers[48]. As a diffraction-based imaging technique, Bragg interferometry is also distinctively compatible with both freely suspended and encapsulated moiré structures. Leveraging this capability, we found that hBN capping layers suppress out-of-plane relaxation modes and subsequently promote in-plane deformations, implicating critical connections between sample design, substrate effects, and emergent properties in moiré systems. The versatility of this methodology could also enable extension to more complex, multi-component vertical heterostructures, such as those containing gate electrodes, opening

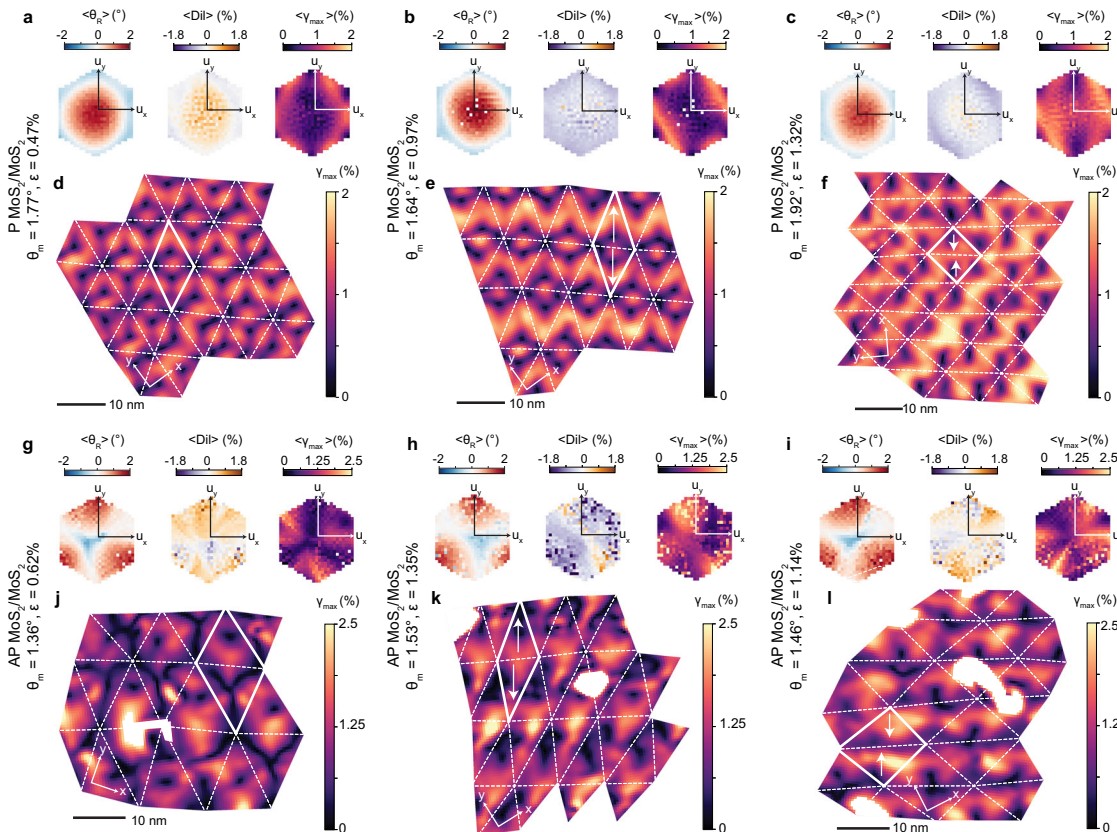

**Fig. 5 | Effects of heterostrain in twisted homobilayers.** Average reconstruction rotation ($\langle\theta_R\rangle$), dilation ($\langle Dil\rangle$), and maximum shear strain ($\langle\gamma_{max}\rangle$) as a function of interlayer displacement ($u_x, u_y$) for P (**a**–**c**) and AP (**g**–**i**) MoS$_2$/MoS$_2$ moiré homobilayers with varying amounts of heterostrain. Corresponding maps of maximum shear strain ($\gamma_{max}$) shown in **d**–**f** and **j**–**l**. The overlaid dashed lines correspond to the moiré unit cell geometry, determined from the displacement maps. White arrows indicate moiré unit cell extension (**e**, **k**) or compression (**f**, **l**). Scan regions affected by sample charging during data acquisition have been removed for clarity in **j**–**l**.

avenues for direct correlative measurements between lattice reconstruction and electrically controllable emergent (opto)electronic phenomena.

## Methods

### Electron microscopy measurements

Electron microscopy was performed at the National Center for Electron Microscopy in the Molecular Foundry at Lawrence Berkeley National Laboratory. Four-dimensional STEM datasets were acquired using a Gatan K3 direct detection camera located at the end of a Gatan Continuum imaging filter on a TEAM I microscope (aberration-corrected Thermo Fisher Scientific Titan 80–300). The microscope was operated in energy-filtered STEM mode at 80 kV with a 10 eV energy filter centred around the zero-loss peak, an indicated convergence angle of 1.71 mrad, and a typical beam current of 45–65 pA depending on the sample. These conditions yield an effective probe size of 1.25 nm (full-width at half-maximum value). Diffraction patterns were collected using a step size of either 0.5 nm or 1 nm with 50 × 50 to 300 × 300 beam positions, covering an area ranging from 25 nm x 25 nm to 300 nm x 300 nm. The K3 camera was used in full-frame electron counting mode with a binning of 4 × 4 pixels and an energy-filtered STEM camera length of 800 mm. Each diffraction pattern had an exposure time of 13 ms, which is the sum of multiple counted frames.

### Displacement fitting overview

The local in-plane interlayer displacement vectors **u**, shown in Fig. 1d, e, were extracted from the 4D-STEM datasets following a procedure generalized from previous work[25]. To summarize, for each diffraction pattern (associated with an individual real space pixel) the average diffuse scattering was first fit to a Lorentzian profile and removed. The average intensities in each of the twelve regions of Bragg disk overlap ($I_j$, Fig. 1b, c) were then fit to the trigonometric expression given in Eq. 1, derived using the weak phase object approximation, to determine the local **u**. Derivation of the fitting function and details of the iterative fit procedure used are outlined in Supplementary Notes 2 and 3. In this equation and all subsequent analysis, $\mathbf{g}_j$ are the reciprocal space vectors associated with each Bragg peak, $\mathbf{a}_1$ and $\mathbf{a}_2$ are the average real space lattice vectors from the two TMD layers, and $A_j, B_j, C_j$ are constants that we fit.

We note that adding integer multiples of $\mathbf{a}_1$ and $\mathbf{a}_2$ to **u** results in the same local diffraction pattern intensities $I_j$, as expected from the system's translational symmetry. Furthermore, flipping the sign of **u** results in the same local diffraction pattern in parallel-stacked materials, for which $B_j \approx 0$. Owing to these ambiguities, the raw displacement vectors obtained are not smoothly oriented and cannot be naïvely differentiated. To combat this, we determine the sign of **u** and the lattice vector offsets needed to obtain smoothly varying displacements following an unwrapping procedure outlined in Supplementary Note 4. This procedure involves using one of several strategies to partition the displacements into zones of constant lattice vector offsets ($n, m$) such that $\mathbf{u} + n\mathbf{a}_1 + m\mathbf{a}_2$ is continuously oriented, followed by use of a mixed-integer program to refine zone boundaries.

### Strain mapping

To obtain the strain maps presented in Figs. 2–5, the zone-unwrapped displacement field **u** is smoothed with a Gaussian filter ($\sigma = 2$ pixels per

$a_0$ where $a_0$ is the average lattice constant for the two layers) and differentiated numerically using a centered 3-pt finite difference stencil. We then divide the interlayer displacement vector by two to obtain the single intralayer displacement $\mathbf{u}^{top}$. Our analysis therefore assumes that the measured displacements can be equally partitioned between the two layers such that $\mathbf{u}^{top} = (\mathbf{r}^{top} - \mathbf{r}^{bottom})/2$ and the two single layer displacements are equal and opposite. This is enforced by symmetry in the homobilayers and, while it may not in principle hold true for generic heterobilayers, calculations suggest that the relaxation magnitudes in $MoS_2$ and $WSe_2$ differ by only 13% (Supplementary Note 7).

We then rely on the small displacement theory approximation (also known as infinitesimal strain theory) in which the displacement gradient is assumed small enough to motivate a linearization of the strain tensors[42]. Specifically we note that this approximation is valid when $\| \nabla \mathbf{u}^{top} \|_\infty \ll 1$ such that the deformation is infinitesimal[42]. In rigid moiré systems, $\mathbf{u}^{top}$ varies between 0 and $a_0/4$ over a length scale $\lambda/2$ (see Supplementary Fig. 13) such that the local deformation gradient is on the order of $a_0/(2\lambda)$. The following analysis therefore assumes a large moiré pattern $\lambda \gg a_0$ and that the local variation of $\mathbf{u}^{top}$ in reconstructed materials preserves $\| \nabla \mathbf{u}^{top} \|_\infty \ll 1$. Under these assumptions, we can decompose the displacement gradient $\nabla \mathbf{u}^{top}$ as the sum of a symmetric infinitesimal strain tensor $\underline{\epsilon}$ and a skew-symmetric infinitesimal rotation matrix $\underline{\omega}$. This results in the following expression, where we have defined the total local rotation in the top layer $\nabla \times \mathbf{u}^{top}$ as $\theta_T^{top}$.

$$\nabla u^{top} = \begin{bmatrix} \frac{\partial u_x^{top}}{\partial_x} & \frac{\partial u_x^{top}}{\partial_y} \\ \frac{\partial u_y^{top}}{\partial_x} & \frac{\partial u_y^{top}}{\partial_y} \end{bmatrix} = \underbrace{\begin{bmatrix} \frac{\partial u_x^{top}}{\partial_x} & \frac{1}{2}\left(\frac{\partial u_x^{top}}{\partial_y} + \frac{\partial u_y^{top}}{\partial_x}\right) \\ \frac{1}{2}\left(\frac{\partial u_x^{top}}{\partial_y} + \frac{\partial u_y^{top}}{\partial_x}\right) & \frac{\partial u_y^{top}}{\partial_y} \end{bmatrix}}_{\underline{\epsilon}} + \underbrace{\begin{bmatrix} 0 & \frac{1}{2}\theta_T^{top} \\ -\frac{1}{2}\theta_T^{top} & 0 \end{bmatrix}}_{\underline{\omega}} \quad (2)$$

The eigenvalues and eigenvectors of $\underline{\epsilon}$ are termed the principal stretches and their directions, respectively, which represent the deformations in pure stretch. The intralayer dilation, or dilatation, represents the relative variation in volume and is given by the trace of $\underline{\epsilon}$. The maximum intralayer engineering shear strain, $\gamma_{max}$, is the difference between the maximum and minimum eigenvalues of $\underline{\epsilon}$. In order to assess the local rotation and dilation due to reconstruction, we subtracted off the rotation and dilation expected from a rigid moiré with the same twist angle, lattice constant mismatch, and/or heterostrain (see Supplementary Notes 5 and 6 for details). We note that this is possible because the local strain is additive within infinitesimal strain theory[42].

### Rotational calibration

To obtain the strain tensor, we needed to account for the rotational offset between the displacement vector coordinate system and the 4D-STEM scan axes. This rotational offset is controlled by two factors. The first is a rotation between the diffraction pattern and the scan direction inherent to the instrument, which we measured as 191° from comparison of an ADF image and corresponding unscattered beam in the diffraction pattern using a defocused STEM probe. The second rotational offset is one of convenience that arises because we rotated the y-axis to align with the $1\bar{1}00$ overlap region (corresponding to orienting the x-axis along the average of the two layers' real space lattice vectors) in order to simplify the mathematics for the fitting and unwrapping procedures. Once the rotational calibration is complete, samples with moiré patterns controlled only by interlayer twist will have a y-axis oriented along SP1 soliton walls, samples with moiré patterns controlled only by lattice mismatch or heterostrain will have an x-axis oriented along the SP1 solitons, and moirés formed from a combination will have soliton orientations somewhere in between (see Supplementary Note 10), in line with what we observed and used to validate our approach. The relative effect of sample drift on the moiré

pattern is deemed minor following the same argument provided in Ref. 25.

### Stacking area and strain trends

Stacking area percents (Figs. 2l, n, 3a–c) were determined by partitioning the interlayer displacement vectors $\mathbf{u}$ into stacking order categories as described in Supplementary Note 8. Average strain quantities (Fig. 2k, m) were collected by partitioning each real space pixel in the same manner and averaging the desired strain quantity for each set of pixels with the same stacking assignment. Average strain hexagons (Figs. 2a-i, 3d–f,h–j, 4d–i, 5a–c, g–i) were similarly obtained by averaging the strain quantities from all pixels whose displacement vectors were within the same $a_0/25$ by $a_0/25$ bins where $a_0$ is the average lattice constant for the two layers. All statistical analyses and stacking order partitioning were performed after smoothing the interlayer displacements with a Gaussian filter, for which we used a $\sigma = 2$ pixels per $a_0$ for all data included in Figs 2k–n, 3a–c. We note that the Gaussian smoothing, finite difference stencil, and finite width of the electron beam may soften the observed strain and stacking features but do not affect the overall trends or conclusions drawn.

### Computational implementation

All processing and analyses of 4D-STEM data were performed using Python on a personal computer, using published modules for Bragg disk detection, image processing and optimization[49–53]. All other code was custom-written by the authors. Density functional theory calculations were performed using the Vienna ab initio software package (VASP)[54] with further calculation parameters given in Supplementary Note 7.

### Data availability

The data supporting the findings of this study are publicly available on Zenodo[55].

### Code availability

The code used for data processing and strain analysis is publicly available on GitHub via Zenodo[56].

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

## Acknowledgements

This material is based upon work supported by the U.S. National Science Foundation (NSF) under award no. DMR-2238196 (D.K.B.). I.M.C. acknowledges support from a University of California, Berkeley Berkeley Fellowship and a National Defense Science and Engineering Graduate (NDSEG) Fellowship under contract FA9550-21-F-0003 sponsored by the Air Force Research Laboratory (AFRL), the Office of Naval Research (ONR) and the Army Research Office (ARO). S.C. acknowledges support from the National Science Foundation under grant no. OIA-1921199. C.O. acknowledges support from a U.S. Department of Energy (DOE) Early Career Research Award. J.C. acknowledges support from the

Presidential Early Career Award for Scientists and Engineers through the U.S. DOE. Work at the Molecular Foundry, LBNL was supported by the Office of Science, Office of Basic Energy Sciences, the U.S. DOE under Contract no. DE-AC02-05CH11231. Computational studies were carried out using supercomputing resources of the National Energy Research Scientific Computing Center (NERSC) and the TMF clusters managed by the High-Performance Computing Services Group at LBNL and were supported by the Laboratory Directed Research and Development Program of LBNL under the same Contract No. Other instrumentation used in this work was supported by grants from the Gordon and Betty Moore Foundation EPiQS Initiative (Award no. 10637, D.K.B.), Canadian Institute for Advanced Research (CIFAR-Azrieli Global Scholar, Award no. GS21-011, D.K.B.), and the 3M Foundation through the 3M Non-Tenured Faculty Award (no. 67507585, D.K.B.). K.W. and T.T. acknowledge support from the Elemental Strategy Initiative conducted by the Ministry of Education, Culture, Sports, Science and Technology, Japan (grant no. JPMXP0112101001) and Japan Society for the Promotion of Science, Grants-in-Aid for Scientific Research (KAKENHI; grant nos. 19H05790, 20H00354 and 21H05233).

## Author contributions

M.V.W. and D.K.B. conceived the study. M.V.W. designed and fabricated the samples. M.V.W., K.C.B., and J.C. acquired the 4D-STEM data. I.M.C. created the data analysis code and strain calculation framework with input from C.O. S.C. carried out DFT and relaxation simulations. M.D. performed SHG measurements. T.T. and K.W. provided the bulk hBN crystals. I.M.C. and M.V.W. processed and analyzed the data. M.V.W., I.M.C., and D.K.B. interpreted the data and wrote the manuscript. D.K.B., S.M.G., and A.R. supervised the work. All the authors contributed to the overall scientific discussion and edited the manuscript.

## Competing interests

The authors declare no competing interests.
