## [Peer Review File · Nature Communications]

Rotational and Dilational Reconstruction in Transition Metal Dichalcogenide Moiré BilayersREVIEWER COMMENTS

Reviewer #1 (Remarks to the Author):

The reliable generation of tunable moirés in 2D material heterostructures ignites the intensive study on their novel (opto-)electronic properties, where the vdW interface reconstruction and resulted strain re-distribution matter a lot. In the manuscript, the authors demonstrate a systematic evaluation of the strain fields in both twisted MoS₂ homobilayers and WSe₂/MoS₂ heterobilayers using an adapted 4D STEM method, which has been similarly applied to the twisted graphene for quantifying the strain distribution in reconstructed interfaces (Nat. Mater. 20, 956-963 (2021)). Although the atomic reconstruction in twisted transition metal dichalcogenide homobilayers and heterobilayers (Nat. Nanotechnol. 15, 592-597 (2020)), as well as the resulted strain modulations (Nanoscale, 13, 13624-13630, (2021)) have also been revealed previously using other techniques, I think the direct evidence of distinct reconstruction driving forces reported here, i.e. local rotations for twisted homobilayers and in-plane dilations for heterobilayers, is still important finding for the 2D community. The sample fabrication, data analysis and theoretic modeling are of high quality, and the manuscript is well prepared. Therefore, I suggest a major revision to resolve the following concerns before the paper gets stronger to be published in Nature Communications.

1) In the abstract, the authors state that "Imaging of TMD moirés has so far established a qualitative understanding of lattice relaxation in terms of interlayer stacking energies, and quantification of strain has relied exclusively on theoretical simulations", putting their advances on the 4D STEM quantification of strain fields in TMD moirés, which I think is not the case. As mentioned above, similar 4D STEM method, actually from the same group, has been used to quantify the strain distribution in reconstructed graphene bilayer interfaces (Nat. Mater. 20, 956-963 (2021)). And the complete reconstruction picture in twisted TMD homobilayers and heterobilayers (Nat. Nanotechnol. 15, 592-597 (2020)), as well as the resulted strain modulations (Nanoscale, 13, 13624-13630, (2021)) have also been reported. The significance of current paper in fact lies in the first direct evidence of reconstruction mechanisms of TMD homo- and hetero-bilayers, which remain in some theoretic proposals. The authors should reorganize the abstract and the introduction part to well clarify their originality and novelty.

2) In Figure 3 g and h, only two data points are shown for the relative stacking area and average dilation as a function of twist angles for AP orientation, which are too few statistically to obtain a trend as discussed in page 9. Please include more data points to verify the conclusion.

3) In Figure 4 and paragraphs in page 12, the authors discuss the effects of heterostrain in twisted homobilayers. What is the source of such uniaxial strain? From my experience, different levels of heterostrain can be found in different regions of one sample, the manipulation of heterostrain as discussed in page 14 is thus fancy but of little feasibility. Have the authors tried introducing larger extrinsic strain, such as stressing by an in-situ AFM tip, which is much controllable then? How about the effects of biaxial strain?

4) In page 12, the authors claim that "...implicating hBN encapsulation in attenuating out-of-plane corrugation". But from the sample fabrication section, we can see the TMD bilayers are only one-side capped by BN, one-side free in substrate holes, similar to the situation of topography measurement of apparent corrugation (free top surface and confined bottom on substrate). The effects of BN on reducing corrugation should be verified by comparison to bilayer samples without BN and with two-side encapsulation of BN.

5) Technically, how about the resolution and precision of this interferometric 4D STEM technique, which means how small a spatial feature can be revealed and how small a strain strength variation is required for the effective detection? What is the purpose of a 10-eV energy filter as mentioned in page 15 Methods? Could the authors also comment on the pros and cons of different TEM strain quantification techniques, e.g. atomic electron tomography, geometric phase analysis, precession electron diffraction and their 4D STEM approach? This should benefit the readers interested in the strain characterization but maybe fresh about available choices.

Reviewer #2 (Remarks to the Author):

A review for "Quantitative Image of Intrinsic and Extrinsic Strain in Transition Metal Dichalcogenide Moire Bilayers".

This manuscript studies the relaxation processes in bilayer twisted TMDC systems. Experimentally the work is based on Bragg interferometry – i.e. the interference between overlapping diffraction disks from the two layers. In addition to the experimental work, the manuscript's conclusions rely heavily on complex computational analysis of this data (including simulation).

Overall, to the best of my ability to judge work that covers such a diverse range of topics/techniques, the work seems well done. It certainly seems worth publishing. It seems like an obvious extension of their recent Nature Materials publication which applied the same methods to twisted bilayer graphene. I do have some concerns though that should be considered by the editor/authors prior to publication:

One notable concern I currently have is that the manuscript is unnecessarily challenging to read – this readability should be notably improved prior to publication. For example:

- While a reader may know what "A" and "AP" refer to, it cannot be assumed that the average reader will know – these should be defined on first use.
- Much of the time the text references figure panels in a seemingly random order – and in addition, most figure panels are not actually labelled well. For example, Fig. 2a,b,c are not actually labelled with P/AP titles and so it is difficult to see the patterns. Same with almost all the other subpanels/figures. Right now, the reader must meticulously read figure captions and the text to have any hope to understand the figures.
- As far as I can tell, the three subpanels associated with Fig 2j and the two associated with Fig. 3i are never referenced in the main text. Nor do they have any colormap/legends. All figures should be discussed in the main text.
- As a reader I appreciate seeing the non-averaged data in Fig. 2 a,b,c. And then the author also shows the averaged data as both explicitly labelled (dilation) and unlabeled (rotation) subpanels. Normally, I'd assume that the averaged data would be front and center in the discussion since it would be presumed to be higher quality and easier to interpret due to the averaging. But instead, the averaged data is relinquished to a 1 cm wide subpanel, and the data itself seems noisier than the unaveraged data -- and additionally, the averaged data is only mentioned as an afterthought in the text. This all makes it very challenging for a reader to understand what the authors are trying to communicate.
- The entire manuscript relies on understanding displacements and the authors use the XM, MX, SP, MMXX, etc. notations without any discussion. The authors should at least cursorily discuss these notations and reference the nice diagrams they made in Fig 1.

Some additional comments:

- This manuscript, as is the case for most 4D STEM manuscripts, relies on a giant dataset that can't be presented in raw form in a manuscript. Instead, the hypercube of data is reduced to simple maps/plots that are human interpretable. The problem I see with this particular manuscript is that the algorithms and analysis are more complex than is typical, and they are not described in detail in a 'methods' style publication already and so they are a bit more 'black box' (even accounting for the supplementary materials) than is standard. The authors state that code and data is available 'on request', but I'd like to see the code and at least one example dataset published alongside the manuscript. To be blunt, as a reviewer, I am forced to simply trust that they did all their complex analysis correctly without typos/errors. But an interested reader should be able to have direct access to sufficient data/code (without explicitly requesting it) to verify this in detail.
- As noted above, the averaged maps of theta_R and Dil seem far noisier than the raw maps would indicate. What is going on here?
- The maps all look "nice" to the eye and so it makes the reader feel like the fitting procedure for determining I_j , A_j , B_j , C_j and u is behaving itself. But given the significant data analysis procedure it is challenging to have a good indication on how unique and accurately the u vectors are determined in the application of the I_j equation. Or phrased differently, how can the authors assure the reader that the set of I_j equations uniquely determine the displacement vector? Under what conditions is this the case and are there any conditions under which this is not the case?
- Much is made about the quantitative nature of the measurements. Can the authors comment on how transferrable they expect these measurements are to non-encapsulated TMDCs? Or TMDCs placed on substrates?

- Why are the FoV's so limited? Is it debris and contamination?
- Why was hBN encapsulation preferred?

Reviewer #3 (Remarks to the Author):

The article presents an extension to a previously published interferometric technique allowing its application to twisted 2D material systems without inversion centres and where the layers have different structures - greatly increasing its potential usefulness. The authors demonstrate the technique by calculating strain from measured lattice displacements in rotated MoS₂ homobilayers and aligned MoS₂/WSe₂ heterobilayers, which has previously been estimated only from the shape of the domains. Notably, the presented technique is able to resolve details of the reconstruction pattern in hBN supported and encapsulated samples meaning that it could be applied to conventional transport devices and would be possible to measure atomic reconstruction under applied fields [e.g. Weston et al, Nature Nano 17 390].

Details of the strain distribution in the samples agree with those assumed in the literature. The hint that the effective strain distribution is affected by the presence of the hBN support (fig 3 j,k and discussion) is interesting but not explored further.

The conclusions within the work are sound and well supported by the experimental results. The presented technique and the complex fitting routine are well explained in the supplementary information. It would be sufficient to approximate the technique, but making the fitting code publicly available before publication would aid replication.

Overall, the paper is extremely well written and presented and the presented technique will be of great use to many research groups. The paper is suitable for publication in nature communications.

I have only one query - I am slightly concerned about the equal partitioning of the displacements between the layers when one is hBN supported (which is not accounted for in the relaxation simulations). Could the authors comment on this? A study of dependence of the relaxation on whether the hBN is present on one or both sides of the system would be interesting but obviously beyond the scope of the current paper.

We are very grateful for the helpful feedback received from the reviewers on our manuscript. We have carefully addressed the reviewer's comments and prepared a revised manuscript. The accompanying revisions to the text are detailed in the responses to the reviewers' individual points in the pages that follow, and we provide two versions of the revised manuscript and SI with one set that is highlighted for clarity (pink for changes in response to Reviewer 1, Blue for changes in response to Reviewer 2, Yellow for changes in response to all Reviewers).

We have undertaken additional experiments and made several changes to the manuscript in response to the reviewers' feedback. In summary, the *major* revisions we have made are:

- Changed title, reorganized abstract and second paragraph of introduction to emphasize novelty of results (i.e. experimental evidence of reconstruction mechanics in TMDs)
- Redesigned Figure 3 and updated corresponding discussion to highlight P vs AP heterobilayer reconstruction and small vs large twist angle reconstruction
- Prepared new heterobilayer samples and performed additional experiments for studies of hBN encapsulation effects, resulting in a new figures (Fig. 4, SI Fig. 15, SI Fig. 16) and section of text ("Effects of Encapsulation Layers on Reconstruction")
- Added Supplementary Section 11 discussing uncertainty in our measurements
- Published data sets used in the manuscript on Zenodo as well as analysis code on Github
- Added labels throughout Figs. 2, 3, and 5 (previously Fig. 4) to improve readability
- Reorganized sections "Rotational reconstruction in homobilayers" and "Dilational reconstruction in heterobilayers" so that figure panels are referenced in order

We think that the manuscript has been substantially improved by these revisions and thank the referees for their thoughtful feedback.

Reviewer #1 (Remarks to the Author):

The reliable generation of tunable moirés in 2D material heterostructures ignites the intensive study on their novel (opto-)electronic properties, where the vdW interface reconstruction and resulted strain re-distribution matter a lot. In the manuscript, the authors demonstrate a systematic evaluation of the strain fields in both twisted MoS₂ homobilayers and WSe₂/MoS₂ heterobilayers using an adapted 4D STEM method, which has been similarly applied to the twisted graphene for quantifying the strain distribution in reconstructed interfaces (Nat. Mater. 20, 956-963 (2021)). Although the atomic reconstruction in twisted transition metal dichalcogenide homobilayers and heterobilayers (Nat. Nanotechnol. 15, 592-597 (2020)), as well as the resulted strain modulations (Nanoscale, 13, 13624-13630, (2021)) have also been revealed previously using other techniques, I think the direct evidence of distinct reconstruction driving forces reported here, i.e. local rotations for twisted homobilayers and in-plane dilations for heterobilayers, is still important finding for the 2D community. The sample fabrication, data analysis and theoretic modeling are of high quality, and the manuscript is well prepared. Therefore, I suggest a major revision to resolve the following concerns before the paper gets stronger to be published in Nature Communications.

We thank the reviewer for their positive comments about the quality and importance of our work.

1) In the abstract, the authors state that “Imaging of TMD moirés has so far established a qualitative understanding of lattice relaxation in terms of interlayer stacking energies, and quantification of strain has relied exclusively on theoretical simulations”, putting their advances on the 4D STEM quantification of strain fields in TMD moirés, which I think is not the case. As mentioned above, similar 4D STEM method, actually from the same group, has been used to quantify the strain distribution in reconstructed graphene bilayer interfaces (Nat. Mater. 20, 956-963 (2021)). And the complete reconstruction picture in twisted TMD homobilayers and heterobilayers (Nat. Nanotechnol. 15, 592-597 (2020)), as well as the resulted strain modulations (Nanoscale, 13, 13624-13630, (2021)) have also been reported. The significance of current paper in fact lies in the first direct evidence of reconstruction mechanisms of TMD homo- and hetero-bilayers, which remain in some theoretic proposals. The authors should reorganize the abstract and the introduction part to well clarify their originality and novelty.

We thank the reviewer for their suggestion. We have reorganized the abstract, introduction, and title of the manuscript to clarify and emphasize the novelty of our results.

2) In Figure 3 g and h, only two data points are shown for the relative stacking area and average dilation as a function of twist angles for AP orientation, which are too few statistically to obtain a trend as discussed in page 9. Please include more data points to verify the conclusion.

We thank the reviewer for their comment. It is challenging to make heterobilayer samples with a specific target twist angle since they are prepared by manually aligning the straight edges of two separate monolayers. While we have made 10+ heterobilayer samples for these studies, their twist angles have been clustered near 0° , 1° , or $2-3^\circ$. As a result, we do not have more intermediate data points in the $0-1^\circ$ twist range. As an alternative, we have modified Fig. 3 and the corresponding text to focus on comparison of heterobilayer reconstruction in large ($\sim 1^\circ$) versus small ($\sim 0^\circ$) twist and have removed any suggestions of finer trends. The revised figure 3 is shown below for convenience.

3) In Figure 4 and paragraphs in page 12, the authors discuss the effects of heterostrain in twisted homobilayers. What is the source of such uniaxial strain? From my experience, different levels of heterostrain can be found in different regions of one sample, the manipulation of heterostrain as discussed in page 14 is thus fancy but of little feasibility. Have the authors tried introducing larger extrinsic strain, such as stressing by an in-situ AFM tip, which is much controllable then? How about the effects of biaxial strain?

For the samples shown in Figure 4 (now Figure 5 in the revised manuscript) the source of uniaxial heterostrain is from one layer stretching slightly relative to the other during the sample fabrication process. As the reviewer has noted, this form of heterostrain from sample fabrication is not

typically deliberate and manifests randomly throughout a sample. However, we are not claiming that heterostrain can be manipulated during sample preparation. Rather, based on our observations that heterostrain substantially affects strain field distributions, we are suggesting that controlled application of a mechanical strain, perhaps with a commercial “push-to-pull” TEM substrate or flexible substrate or other nanofabrication approaches being developed in the field, should allow dynamic tuning of reconstruction strain patterns. While intentional introduction of uniaxial or biaxial strain would be quite interesting, such an experiment is well beyond the scope of our current work and therefore we have not attempted this yet.

To clarify our point we have added the following text to the conclusion:

“Since intralayer strain affects the positions of the conduction and valence band edges throughout the moiré unit cell and can generate an in-plane piezopotential, manipulating the arrangement and magnitude of reconstruction strain via **extrinsic application of uniaxial heterostrain** should have important implications for changing the moiré potential landscape and localization of charge carriers.”

4) In page 12, the authors claim that “...implicating hBN encapsulation in attenuating out-of-plane corrugation”. But from the sample fabrication section, we can see the TMD bilayers are only one-side capped by BN, one-side free in substrate holes, similar to the situation of topography measurement of apparent corrugation (free top surface and confined bottom on substrate). The effects of BN on reducing corrugation should be verified by comparison to bilayer samples without BN and with two-side encapsulation of BN.

The reviewer raises a good point. To investigate this, we fabricated more samples and performed additional experiments on AP heterobilayer samples with three regions: hBN on top *and* bottom, hBN on top *alone*, and *no* hBN (i.e., fully suspended). The information gathered from this study is now included as a new figure (Fig. 4), which is shown below. By comparing the dilations measured in these different regions with varying extents of encapsulation, we now provide direct evidence that out-of-plane corrugations are almost entirely suppressed in the fully encapsulated region and partially suppressed in the partially encapsulated region, as shown in Fig. 4k.

5) Technically, how about the resolution and precision of this interferometric 4D STEM technique, which means how small a spatial feature can be revealed and how small a strain strength variation is required for the effective detection?

As noted in the methods section, the full-width at half maximum of the electron beam is ~ 1.25 nm for the chosen imaging parameters. Based on this, our spatial resolution is on the order of 1 nm.

Regarding the strain uncertainty and detection limit, we have now added a section in the supplementary information (Supplementary Section 11) in which we present the residuals and RMSE associated with the fitting and unwrapping procedures. The resultant strain detection limit from this analysis is on the order of a tenth of a degree rotation and a tenth of a percent dilation for the procedure used.

What is the purpose of a 10-eV energy filter as mentioned in page 15 Methods?

The energy filter reduces background noise from inelastic scattering in the diffraction patterns.

Could the authors also comment on the pros and cons of different TEM strain quantification techniques, e.g. atomic electron tomography, geometric phase analysis, precession electron diffraction and their 4D STEM approach? This should benefit the readers interested in the strain characterization but maybe fresh about available choices.

We thank the reviewer for their suggestion. We have added a comment to the manuscript describing why we are using intensity information to calculate strain as opposed to diffraction disk positions, which is the most similar strain mapping method available:

"While strain mapping of 2D materials has typically been performed by tracking changes in Bragg disk positions throughout a 4D-STEM data set,^{38,39} accurate registration of disk positions is very challenging for moiré systems where diffraction disks from the constituent monolayers have substantial overlap and a relatively low signal-to-noise ratio. As an alternative, by taking the gradient of the displacement vector field, we can calculate the intralayer 2D strain tensor at each position in the sample.^{25,40–42}"

Reviewer #2 (Remarks to the Author):

A review for “Quantitative Image of Intrinsic and Extrinsic Strain in Transition Metal Dichalcogenide Moire Bilayers”.

This manuscript studies the relaxation processes in bilayer twisted TMDC systems. Experimentally the work is based on Bragg interferometry – i.e. the interference between overlapping diffraction disks from the two layers. In addition to the experimental work, the manuscript’s conclusions rely heavily on complex computational analysis of this data (including simulation).

Overall, to the best of my ability to judge work that covers such a diverse range of topics/techniques, the work seems well done. It certainly seems worth publishing. It seems like an obvious extension of their recent Nature Materials publication which applied the same methods to twisted bilayer graphene.

We thank the reviewer for their positive feedback toward our work.

I do have some concerns though that should be considered by the editor/authors prior to publication:

One notable concern I currently have is that the manuscript is unnecessarily challenging to read – this readability should be notably improved prior to publication. For example:

- While a reader may know what “A” and “AP” refer to, it cannot be assumed that the average reader will know – these should be defined on first use.

We thank the reviewer for bringing this issue to our attention. We have now defined P and AP on first use in the introduction:

“parallel (P, 3R-like, near 0°) or anti-parallel (AP, 2H-like, near 60°)”

- Much of the time the text references figure panels in a seemingly random order – and in addition, most figure panels are not actually labelled well. For example. Fig. 2a,b,c are not actually labelled with P/AP titles and so it is difficult to see the patterns. Same with almost all the other subpanels/figures. Right now, the reader must meticulously read figure captions and the text to have any hope to understand the figures.

We thank the reviewer for their comment and apologize for any confusion. Additional labels have been added throughout Figs. 2–5 to ease readability. We have also modified both the text and figures so that the figure panels are referenced in order. One exception remains in Fig. 5, where

we feel reorganizing the figure panels to match the order of the narrative or vice versa would introduce more confusion. In this case, we have left the figure layout as is in order to keep certain sets of related information grouped together.

- As far as I can tell, the three subpanels associated with Fig 2j and the two associated with Fig. 3i are never referenced in the main text. Nor do they have any colormap/legends. All figures should be discussed in the main text.

We have now added color scales to these figure panels. We have also added references to these figure panels in the text as follows:

“Schematics depicting the rotation-driven reconstruction model and accumulation of shear strain at domain boundaries are provided in Fig. 2j.”

“Altogether, these measurements show that the lattice constant mismatch decreases (increases) in domains with low (high) stacking energy to cause local volumetric expansion (contraction) of these domains, as depicted in the schematics in Fig. 3g.”

- As a reader I appreciate seeing the non-averaged data in Fig. 2 a,b,c. And then the author also shows the averaged data as both explicitly labelled (dilation) and unlabeled (rotation) subpanels. Normally, I'd assume that the averaged data would be front and center in the discussion since it would be presumed to be higher quality and easier to interpret due to the averaging. But instead, the averaged data is relinquished to a 1 cm wide subpanel, and the data itself seems noisier than the unaveraged data -- and additionally, the averaged data is only mentioned as an afterthought in the text. This all makes it very challenging for a reader to understand what the authors are trying to communicate.

The reviewer raises a good question. In a rigid moiré, all points in the displacement space would be equally sampled; however, reconstruction affects this distribution and causes some parts of the displacement space to become more populated than others. Therefore the reason that the averaged quantities look deceptively noisier than the single measurements is that the averaged plots show the rotation/dilation values at all possible displacement vector values, but some of these displacement vectors are infrequently sampled in real-space. However, we think it is insightful to include both the unaveraged real-space maps, which show the periodic nature of the deformations, as well as the averaged plots, which explicitly correlate the measured rotations/dilations to their displacement values.

- The entire manuscript relies on understanding displacements and the authors use the XM, MX, SP, MMXX, etc. notations without any discussion. The authors should at least cursorily discuss these notations and reference the nice diagrams they made in Fig 1.

We thank the reviewer for pointing out that the notation used is not currently clear. We have noted the different stacking orders to the main text as follows:

“Example displacement maps for P and AP moiré bilayers are provided in Fig. 1c,d. For the P case, the high-symmetry stacking orders include MMXX, XM, MX, and SP (saddle point), as described in Fig. 1e. In comparison, the stacking orders in the AP case include XMMX, MM, XX, and SP, shown in Fig. 1f.”

We have also added text to the caption for Fig. 1f, g to clarify:

“(e,f) High-symmetry stacking sequences and corresponding displacement vectors for P and AP H-phase TMD moiré bilayers. Saddle points, or domain walls, abbreviated as SP. Metal and chalcogen denoted as M and X, respectively.”

Some additional comments:

- This manuscript, as is the case for most 4D STEM manuscripts, relies on a giant dataset that can't be presented in raw form in a manuscript. Instead, the hypercube of data is reduced to simple maps/plots that are human interpretable. The problem I see with this particular manuscript is that the algorithms and analysis are more complex than is typical, and they are not described in detail in a 'methods' style publication already and so they are a bit more 'black box' (even accounting for the supplementary materials) than is standard. The authors state that code and data is available 'on request', but I'd like to see the code and at least one example dataset published alongside the manuscript. To be blunt, as a reviewer, I am forced to simply trust that they did all their complex analysis correctly without typos/errors. But an interested reader should be able to have direct access to sufficient data/code (without explicitly requesting it) to verify this in detail.

We thank the reviewer for their comment. Indeed, we always intended to publish the full data analysis code and all data sets used upon publication, as is our custom. The code is publicly available at: <https://github.com/bediakolab/pyInterferometry> and the datasets are available at <https://doi.org/10.5281/zenodo.7779105>.

- As noted above, the averaged maps of theta_R and Dil seem far noisier than the raw maps would indicate. What is going on here?

We thank the reviewer for their question. We address this concern in our response to the reviewer's previous questions about the averaged versus non-averaged data.

- The maps all look “nice” to the eye and so it makes the reader feel like the fitting procedure for determining I_j , A_j , B_j , C_j and u is behaving itself. But given the significant data analysis procedure it is challenging to have a good indication on how unique and accurately the u vectors

are determined in the application of the I_j equation. Or phrased differently, how can the authors assure the reader that the set of I_j equations uniquely determine the displacement vector? Under what conditions is this the case and are there any conditions under which this is not the case?

The reviewer raises an important question. The displacement unwrapping process, which is currently described in the methods section and Supplementary Section 4, directly addresses this concern :

“We note that adding integer multiples of a_1 and a_2 to u results in the same local diffraction pattern intensities I_j , as expected from the system’s translational symmetry. Furthermore, flipping the sign of u results in the same local diffraction pattern in parallel-stacked materials, for which $B_j \approx 0$. Owing to these ambiguities, the raw displacement vectors obtained are not smoothly oriented and cannot be naïvely differentiated. To combat this, we determine the sign of u and the lattice vector offsets needed to obtain smoothly varying displacements following an unwrapping procedure outlined in Supplementary Section 4. This procedure involves using one of several strategies to partition the displacements into zones of constant lattice vector offsets (n, m) such that $u + na_1 + ma_2$ is continuously oriented, followed by use of a mixed-integer program to refine zone boundaries.”

- Much is made about the quantitative nature of the measurements. Can the authors comment on how transferrable they expect these measurements are to non-encapsulated TMDCs? Or TMDCs placed on substrates?

The reviewer poses an important question. To investigate this, we performed additional experiments on AP heterobilayer samples with three regions: hBN on top and bottom (similar to placement on a substrate), hBN on top, and no hBN (i.e., fully suspended). The information gathered from this study is now included as a new figure (Fig. 4, which is shown above in response to a similar question from Reviewer #1) and demonstrates how both in-plane and out-of-plane reconstruction change with the extent of encapsulation.

- Why are the FoV’s so limited? Is it debris and contamination?

The four-dimensional datasets we collect are very large (10’s of GB) and therefore we are limited to a scan window of $\sim 300 \times 300$ pixels. Given our chosen step sizes (typically 0.5 nm), the FOV is then limited to 150×150 nm at most. Most of the time the FOV is smaller than that, again to reduce file sizes and data processing time. Representative microscale TEM images of samples are provided in Section 1 of the Supplementary Info.

- Why was hBN encapsulation preferred?

Encapsulation with hBN is preferred to stabilize the TMD layers when they are suspended over the holes in the TEM grid. Even more fundamentally, the overwhelming majority of (if not all) devices that are fabricated for careful optical/transport measurements of correlated electron physics, moiré excitons, etc. necessitate encapsulation by hBN since this facilitates ultraclean sample fabrication and helps to screen the TMD layers from charge puddles in the SiO₂/Si substrate. So it is essential to probe the structure of these TMD moirés under sample conditions that are representative of the actual physical measurements. As we show in this paper, the relaxation mechanics are indeed affected by encapsulation in a manner that has not been previously appreciated (see the new Figure 4). Encapsulated vs non-encapsulated samples are thus likely to exhibit different physics (given how sensitive these exotic physical phenomena are to structural perturbations of the moiré lattice).

Reviewer #3 (Remarks to the Author):

The article presents an extension to a previously published interferometric technique allowing its application to twisted 2D material systems without inversion centres and where the layers have different structures - greatly increasing its potential usefulness. The authors demonstrate the technique by calculating strain from measured lattice displacements in rotated MoS₂ homobilayers and aligned MoS₂/WSe₂ heterobilayers, which has previously been estimated only from the shape of the domains. Notably, the presented technique is able to resolve details of the reconstruction pattern in hBN supported and encapsulated samples meaning that it could be applied to conventional transport devices and would be possible to measure atomic reconstruction under applied fields [e.g. Weston et al, Nature Nano 17 390].

Details of the strain distribution in the samples agree with those assumed in the literature. The hint that the effective strain distribution is affected by the presence of the hBN support (fig 3 j,k and discussion) is interesting but not explored further.

The conclusions within the work are sound and well supported by the experimental results. The presented technique and the complex fitting routine are well explained in the supplementary information. It would be sufficient to approximate the technique, but making the fitting code publicly available before publication would aid replication.

We thank the reviewer for their comment. We intend to make the analysis code and datasets publicly available upon publication of the manuscript. Also, we have now studied the effect of hBN in more detail.

Overall, the paper is extremely well written and presented and the presented technique will be of great use to many research groups. The paper is suitable for publication in nature communications.

We thank the reviewer for their positive feedback to our work.

I have only one query - I am slightly concerned about the equal partitioning of the displacements between the layers when one is hBN supported (which is not accounted for in the relaxation simulations). Could the authors comment on this? A study of dependence of the relaxation on whether the hBN is present on one or both sides of the system would be interesting but obviously beyond the scope of the current paper.

The reviewer raises a good point. To investigate this, we performed additional experiments on AP heterobilayer samples with three regions: hBN on top and bottom, hBN on top, and no hBN. The information gathered from this study is now included as a new figure (Fig. 4) and demonstrates how encapsulation with hBN affects both in-plane and out-of-plane reconstruction.

REVIEWERS' COMMENTS

Reviewer #1 (Remarks to the Author):

In the response letter and revised documents, the authors addressed my concerns well. I recommend the publication now.

Reviewer #2 (Remarks to the Author):

The authors have edited the manuscript satisfactorily and it is in good condition for publication. Indeed, I see that they have even performed new experiments at the behest of another reviewer and included that into the manuscript.

The authors have noted that the code necessary for the analysis in this manuscript is available on github. For what it is worth the code base is around 8500 lines of code and I view the publication/sharing of that code base (as they have done) as being just as important as the manuscript itself.

Finally I note that the authors have provided the data in an intermediate form on zenodo (correct me if I am wrong), but I have still be unable to locate a 'h5' or 'dm4' file that would represent a raw dataset (i.e. the data dumped from the detector). I will not insist on the authors uploading a raw dataset as the code is at least available now, but I will *****strongly***** encourage the authors to upload at least one (possibly small ROI) dataset to allow a reader to perform a full test of their code. I have found in my work that having a fully functioning code example that goes from raw data to finished analysis is critical for a full understanding of how someone's code/algorithm works. The more complicated the analysis and the more interesting the results -- the more I want this to be given. (If these data are in fact already available then please just make that more apparent to a reader and I apologize for the misunderstanding.)

Reviewer #3 (Remarks to the Author):

The additions and changes have significantly improved the manuscript. In particular the rearrangement of the figures has made it simpler to understand the presented information without careful reading. The additional data on the dependence of the reconstruction on hBN encapsulation is welcome.

The authors should clarify the thickness of the encapsulation layers used, at least approximately - are they thick enough to be considered rigid?

Once this is cleared up the paper is suitable for publication.

Reviewer #1 (Remarks to the Author):

In the response letter and revised documents, the authors addressed my concerns well. I recommend the publication now.

We thank the reviewer for their positive response to our work.

Reviewer #2 (Remarks to the Author):

The authors have edited the manuscript satisfactorily and it is in good condition for publication. Indeed, I see that they have even performed new experiments at the behest of another reviewer and included that into the manuscript.

The authors have noted that the code necessary for the analysis in this manuscript is available on github. For what it is worth the code base is around 8500 lines of code and I view the publication/sharing of that code base (as they have done) as being just as important as the manuscript itself.

Finally I note that the authors have provided the data in an intermediate form on zenodo (correct me if I am wrong), but I have still be unable to locate a 'h5' or 'dm4' file that would represent a raw dataset (i.e. the data dumped from the detector). I will not insist on the authors uploading a raw dataset as the code is at least available now, but I will *****strongly***** encourage the authors to upload at least one (possibly small ROI) dataset to allow a reader to perform a full test of their code. I have found in my work that having a fully functioning code example that goes from raw data to finished analysis is critical for a full understanding of how someone's code/algorithm works. The more complicated the analysis and the more interesting the results -- the more I want this to be given. (If these data are in fact already available then please just make that more apparent to a reader and I apologize for the misunderstanding.)

We thank the reviewer for their feedback. We have now added an example raw dataset to the Zenodo repository for readers to use.

Reviewer #3 (Remarks to the Author):

The additions and changes have significantly improved the manuscript. In particular the rearrangement of the figures has made it simpler to understand the presented information without careful reading. The additional data on the dependence of the reconstruction on hBN encapsulation is welcome.

The authors should clarify the thickness of the encapsulation layers used, at least approximately - are they thick enough to be considered rigid?

Once this is cleared up the paper is suitable for publication.

We thank the reviewer for their comment. We agree that the thickness of the encapsulation layers is an important detail and apologize that it is somewhat hidden in the manuscript. At the beginning of the Interlayer Displacement Mapping section and in the Sample Preparation section

of the SI we note that the hBN used was $\sim 5\text{--}10$ nm thick, which corresponds to roughly 10–20 layers. Based on studies showing the how bending stiffness of vdW materials increases with thickness (Wang, G. et al. *Phys. Rev. Lett.* **2019**, *123*, 116101), we think it is reasonable to assert the hBN in the 5–10 nm thick range is rigid enough to substantially affect corrugation but is not as rigid as, perhaps, a 20-30 nm flake. This could explain why we still see evidence of some out-of-plane relaxation in our fully encapsulated samples, albeit much less than in the samples with less encapsulation.